# The Health Behaviour of German Outpatient Caregivers in Relation to the COVID-19 Pandemic: A Mixed-Methods Study

**DOI:** 10.3390/ijerph18158213

**Published:** 2021-08-03

**Authors:** Natascha Mojtahedzadeh, Felix Alexander Neumann, Elisabeth Rohwer, Albert Nienhaus, Matthias Augustin, Volker Harth, Birgit-Christiane Zyriax, Stefanie Mache

**Affiliations:** 1Institute for Occupational and Maritime Medicine (ZfAM), University Medical Center Hamburg-Eppendorf (UKE), 20459 Hamburg, Germany; n.mojtahedzadeh@uke.de (N.M.); e.rohwer@uke.de (E.R.); harth@uke.de (V.H.); 2Midwifery Science—Health Services Research and Prevention, Institute for Health Service Research in Dermatology and Nursing (IVDP), University Medical Center Hamburg-Eppendorf (UKE), 20246 Hamburg, Germany; fe.neumann@uke.de (F.A.N.); b.zyriax@uke.de (B.-C.Z.); 3Department of Occupational Medicine, Hazardous Substances and Public Health, Institution for Statutory Accident Insurance and Prevention in the Health and Welfare Services (BGW), 22089 Hamburg, Germany; a.nienhaus@uke.de; 4Competence Center for Epidemiology and Health Services Research for Healthcare Professionals (CVcare), Institute for Health Services Research in Dermatology and Nursing (IVDP), University Medical Center Hamburg-Eppendorf (UKE), 20246 Hamburg, Germany; 5Competence Center for Health Services Research in Vascular Diseases (CVvasc), Institute for Health Services Research in Dermatology and Nursing (IVDP), University Medical Center Hamburg-Eppendorf (UKE), 20246 Hamburg, Germany; m.augustin@uke.de

**Keywords:** COVID-19, pandemic, health behaviour, outpatient care, occupational health, eating behaviour, physical activity, ambulatory care, lifestyle

## Abstract

The COVID-19 pandemic has affected outpatient caregivers in a particular way. While the German population becomes increasingly older, the number of people in need of care has also increased. The health and, thus, the health behaviour of employees in the outpatient care become relevant to maintain working capacity and performance in the long term. The aims of the study were (1) to examine the health behaviour and (2) to explore pandemic-related perceived change of health behaviour among outpatient caregivers during the COVID-19 pandemic. In a mixed-methods study, 15 problem-centred interviews and a web-based cross-sectional survey (*N* = 171) were conducted with outpatient caregivers working in Northern Germany. Interviewees reported partially poorer eating behaviour, higher coffee consumption, lower physical activity, skipping breaks more often and less sleep duration and quality during the pandemic. Some quantitative findings indicate the same tendencies. A majority of participants were smokers and reported higher stress perception due to the pandemic. Preventive behaviour, such as wearing PPE or hand hygiene, was increased among interviewees compared to the pre-pandemic period. Our findings indicate that the COVID-19 pandemic could negatively affect outpatient caregivers’ health behaviour, e.g., eating/drinking behaviour and physical activity. Therefore, employers in outpatient care should develop workplace health promotion measures to support their employees in conducting more health-promoting behaviours during the COVID-19 pandemic.

## 1. Introduction

### 1.1. Background

At the end of 2017, there were 3.41 million people in need of care in Germany. In this context, there has been an increase of people taken care of in outpatient care (2017: 829,958, >20% increase compared to 2016) [1] because it allows patients to remain in their home environment as long as possible compared to inpatient settings [2]. The outpatient care service, thus, supports those in need of care and their relatives in caring for them at home. It offers families support and help in everyday life so that caring relatives can better organise their work and care. For example, they carry out body-related care measures, nursing care measures, home nursing care and counselling of those in need of care and their relatives on care-related issues [3]. Despite the increasing need for professional outpatient caregivers, there are shortages of skilled workers and early retirements [4] as well as more days of incapacity for work in care occupations compared to other employee groups in Germany [5]. Several studies on healthcare workers and nurses reported a high prevalence for overweight, obesity and metabolic syndrome [6,7,8] and an association of obesity with longer working hours [9]. Obesity and metabolic syndrome increase the risk of cardiovascular disease and type 2 diabetes mellitus [6,10,11,12] and an inadequate diet and physical inactivity have been reported as the most common antecedents for metabolic syndrome and cardiovascular disease [11]. More profoundly, this association has been demonstrated even for short-term changes in diet and physical activity (e.g., due to home containment) [13]. In addition, the presence of cardiovascular comorbidities, such as obesity and metabolic syndrome, increases the risk that COVID-19 will progress more severely, taking into account that healthcare workers already seem to be at higher risk of infection [14,15,16].

### 1.2. Current State of Research

Several studies have already been published analysing caregivers’ job demands during the COVID-19 pandemic and their strain reactions, e.g., [17,18,19,20,21,22,23,24,25,26,27,28,29,30,31,32,33,34,35]. However, these studies do not include the health behaviour of caregivers at all. A recent publication from Mojtahedzadeh et al. [36] examined the health behaviour (eating behaviour, drinking behaviour, physical activity, smoking behaviour, regeneration and personal health behaviours) of outpatient caregivers from Germany prior to the COVID-19 pandemic [36]. Health behaviour can be divided in positive (e.g., healthy eating) and negative (e.g., consumption of tobacco) behavioural patterns [37,38]. In the following, the current state of research on outpatient caregivers’ health behaviour is presented according to the investigated topics of eating and drinking behaviour, physical activity, smoking behaviour and regeneration behaviour.

#### 1.2.1. Eating and Drinking Behaviour

Various studies on outpatient caregivers and trainees in nursing professions in Germany have indicated that the eating behaviour needs to be improved [8,36,39]. While the literature on fruit and vegetable consumption among nursing and residential care trainees is inconsistent [39,40,41], increased consumption of sweets [39,42] and convenient foods [39] has been reported. In addition, a high caffeine consumption as well as obesity have been described as being a consequence of the working conditions and work-related stress perception in the inpatient care [36,43].

Regarding the COVID-19 pandemic, data from the Robert Koch Institute’s health monitoring of the German population show a significant increase in average body weight (ΔM = +1.1 kg) and average body mass index (ΔM = +0.5 kg/m^2^) for the five-month period after the start of the pandemic from April to August 2020 [44]. A review on changes in eating behaviour of the general population during the COVID-19 pandemic describes more frequent snacking and an increased consumption of high-glycaemic carbohydrate sources which might partially explain the increase of weight [45]. Moreover, an increase in the intake of fruits and vegetables and protein sources, particularly pulses, and a decrease in fresh fish/seafood and alcohol intake were reported. Knowledge about eating behaviour change among inpatient and outpatient caregivers during the pandemic is limited. Among Vietnamese nursing trainees, 42.8% of students reported healthier eating behaviour, while 57.2% reported no change or deterioration [46]; however, no detailed information on food components was supplied.

#### 1.2.2. Physical Activity

An Australian study reported that the majority of inpatient nurses are sufficiently physically active (≥150 min per week) according to the guidelines on physical activity and sedentary behaviour, as recommended by the World Health Organisation [40,47]. However, other nurses from inpatient care are reportedly sufficiently active during working hours [41]. In contrast, alternating night shifts and long working hours have been shown to reduce physical activity of inpatient nurses in leisure time [48], which was also shown among German nursing trainees [8,49]. Furthermore, in an Irish study, it was reported that inpatient nurses of older age (≥40 years) were less likely to comply with physical activity recommendations than their younger colleagues [50]. Along with this, a recent study comparing the physical activity of Polish residents before and during the COVID-19 pandemic found that the frequency of physical activity decreased among males and those aged 39–58 years [51]. It was also reported that a decrease in exercise time was observed in all groups. In addition, a study of Turkish nursing students outlined that 56.7% of participants reported that they did not exercise regularly during home containment [52].

#### 1.2.3. Smoking Behaviour

An international meta-analysis on smoking behaviour of healthcare workers showed that inpatient nurses have the highest prevalence of tobacco use among all hospital occupational groups across all nationalities [53]. The smoking behaviour of outpatient caregivers has so far only been studied in the pre-pandemic context by Mojtahedzadeh, Rohwer, Neumann, Nienhaus, Augustin, Zyriax, Harth and Mache [36]. It is still unknown whether and to what extent their smoking behaviour changed during the COVID-19 pandemic. A comparison of smoking prevalence for the first three months of the pandemic with the same months of the following year based on health reporting data from Germany shows a decrease in smoking prevalence from 32.6% to 28.1% [44]. Smoking has also been described for inpatient nurses as a coping mechanism for dealing with work-related stress during the pandemic [54].

#### 1.2.4. Regeneration Behaviour

Wendsche et al. [55] emphasised that in order to cope with the high workload and time pressure, regular breaks and, thus, important regeneration phases, are avoided. Furthermore, regeneration at night also seems to be impaired among inpatient nurses. Working conditions such as shift work [48] and psychological distress [56] have been reported to reduce sleep quality and duration. Poor sleep quality has interfered with adherence to a healthy lifestyle, such as physical activity [48] and low sleep duration, was associated with higher carbohydrate and lower protein intake in studies of inpatient nurses [57]. With regard to the COVID-19 pandemic, several studies reported inadequate sleep and poor sleep quality among inpatient nurses or nursing trainees [58,59,60]. A pooled prevalence of 43% for sleep disturbance was found among inpatient nurses [61]. Further studies underline a poorer sleeping quality among healthcare workers due to the COVID-19 pandemic [22,62]. A positive correlation between anxiety and sleep quality was reported and female inpatient nurses tended to be more likely to experience sleep disturbances and anxiety than males [58]. However, this is in contrast with Zheng, Wang, Feng, Ye, Zhang and Fan [60], who associated male inpatient nurses in China with poorer sleep quality.

### 1.3. Study Aims and Research Questions

Considering the state of research, the working conditions in outpatient care and the additional burden of the COVID-19 pandemic could be a meaningful factor influencing the health behaviour and health of outpatient caregivers. Therefore, the aim of this mixed-methods study was to examine the health behaviour and potential health behaviour changes of outpatient caregivers during the COVID-19 pandemic in Germany. Thus, a deeper understanding about caregivers’ motives and self-understanding of their health behaviour could be generated. In this study, health behaviour will be described in eating behaviour, physical activity, smoking and regeneration behaviour as well as health-promoting behavioural patterns at work and in private considering the COVID-19 pandemic.

We proposed the following research questions and hypotheses:How do outpatient caregivers perceive their health behaviour during the COVID-19 pandemic?Did outpatient caregivers perceive any change in their health behaviour due to the COVID-19 pandemic?

**Hypothesis** **1.**
*Outpatient caregivers rate the healthiness of their eating behaviour as significantly lower during the pandemic compared to before.*


**Hypothesis** **2.**
*Outpatient caregivers rate their physical activity as lower during the pandemic compared to before.*


**Hypothesis** **3.**
*Outpatient caregivers rate their tobacco use as higher during the pandemic compared to before.*


**Hypothesis** **4.**
*Outpatient caregivers rate their perceived stress as higher during the pandemic compared to before.*


**Hypothesis** **5.**
*Outpatient caregivers rate the quality of their sleep as significantly lower during the pandemic compared to before.*


## 2. Materials and Methods

### 2.1. Study Design

We conducted a sequential mixed-methods study, applying both a qualitative and a quantitative research approach [63]. Firstly, semi-structured interviews were conducted to gain explorative insights of outpatient caregivers in Germany. Subsequently, an online survey to collect quantitative data was carried out. This study was approved by the Local Psychological Ethics Committee of the Hamburg Psychosocial Medical Centre of the University Medical Centre Hamburg-Eppendorf (UKE) (Ethic code: LPEK-0083).

### 2.2. Qualitative Approach

#### 2.2.1. Design, Participants and Recruitment

We conducted 15 problem-centred telephone interviews according to Witzel [64] with outpatient caregivers from Germany in May and June 2020. The method of problem-centred interviews was chosen as it is known to analyse specific behaviour, experiences, reasons, evaluations and subjective opinions in a dialogue and aims at a common understanding process between the interviewer and the interviewee [64,65]. The interviews were carried out by the first author, a Public Health Scientist (M.A.) with prior experiences in qualitative research, working in the field of occupational health psychology. Eligibility criteria included a minimum work experience of six months in outpatient care (since January 2020, prior to the outbreak of the coronavirus in Germany) on a full- or part-time basis. To promote the interview study, we sent emails including project information and study aims to a representative sample of outpatient care services in Northern Germany. In addition, we recruited via social networks (Facebook, Xing and LinkedIn). All outpatient caregivers participated voluntarily and were interviewed only once. None of the interviews were repeated. Prior to the interviews, participants received the study information and signed a written informed consent. Telephone interviews were carried out until data saturation was reached, i.e., no new topics came up during the interviews. Interviews were conducted in German and tape-recorded. The length varied from 26 up to 60 min. Field notes were made immediately after each interview.

#### 2.2.2. Interview Guideline

We developed a semi-structured interview guideline including different topics. In adherence to Misoch [66], the structure of the interview guideline was then divided in four phases. The information phase, the warm-up phase, the main phase and, finally, the final phase or rather end of the interview. An extract of the topic list of the interview guideline is summarised in Appendix A. Further topics (e.g., job demands, resources and strain reactions, occupational health and safety and further needs) are presented elsewhere [67,68].

#### 2.2.3. Data Analysis

All audio recording were transcribed, anonymised and double-checked by the first author. Transcripts of the interviews were analysed by the first author using MAXQDA 2020 (VERBI Software, 2019; VERBI GmbH, Berlin, Germany) for data analysis [69]. A deductive-inductive approach following qualitative content analysis of Mayring [70] was applied. An iterative process followed establishing and refining the used coding system. Several categories as well as subcategories were received and finally summarised in a separate document. The findings of the qualitative study were profoundly discussed within the group of authors. The researchers’ personal involvement and preconceptions were reflected and excluded. Quotes of interviewees were translated from German to English by the first author.

### 2.3. Quantitative Approach

#### 2.3.1. Design, Participants and Recruitment

Between May 2020 and February 2021, we carried out a cross-sectional online survey among 171 outpatient caregivers in Northern Germany. In order to recruit participants, 367 outpatient care companies were contacted by telephone and via the email distribution list of the Hamburg regional group of the “Federal Association of Private Providers of Social Services”. Finally, 253 companies agreed to distribute a flyer with participation information about the study among their employees, whereas 114 companies declined to participate. Reasons given by nursing management varied, but mainly “lack of time”, “lack of interest” and “reluctance to participate in questionnaires” were named. In addition, outpatient caregivers were directly addressed. A total of 607 potential participants visited the online questionnaire homepage, of which 171 (28.2%) started and completed the questionnaire, 315 (51.9%) dropped out and 121 (19.9%) did not participate. The online survey provided information about the purpose, anonymity and voluntariness of the study, and informed consent was obtained. Eligibility criteria were that participants were outpatient caregivers in Northern Germany with at least 6 months of work experience and a working time of at least 25 h per week.

#### 2.3.2. Variables, Measures and Processes

There are no missing data in our dataset, as the data were collected with an appropriately set online tool. Furthermore, the dataset was individually tested for internal validity by two researchers, and data for which there was doubt about the validity were excluded in the process. Unless otherwise noted, statistical analyses were based on a sample size of *N* = 171. The participants’ characteristics collected through the questionnaire included gender, age, height, weight, information on the work situation and the operation area, marital status, cultural origin and educational level. The Body Mass Index (BMI) was defined as BMI = (body weight in kg)/(body height in m)^2^. Overweight was defined as a BMI ≥ 25 kg/m^2^, obesity as BMI ≥ 30 kg/m^2^ [71]. Gender (male, female), age (<40 years, ≥40 years) and BMI (<25 kg/m^2^, ≥25 kg/m^2^) were used as dichotomous grouping variables for statistical analysis. Participants with the gender “diverse” were excluded from the gender-specific analysis, as this group consisted of two persons, which was not sufficient.

Based on randomised-controlled trials, a Mediterranean diet is recommended in European and international guidelines as a “healthy heart” diet [72,73,74,75]. Adherence can be assessed by the 14-item Mediterranean Diet Adherence Screener (MEDAS) [76]. Among others, the MEDAS has been validated for the German population with a fair agreement with the gold standard Food Frequency Questionnaire (FFQ) [73]. MEDAS score and the corresponding score of the FFQ correlated significantly (*r* = 0.52) and showed identical directions for the food group intakes [76]. Thus, the average MEDAS scores show a high degree of agreement to the corresponding FFQ scores and are only slightly higher [73,76]. To describe dietary quality, we used the adapted German version of this instrument [73]. It queries the consumption frequency of foods (olive oil, vegetables, fruit, red meat, animal fats, carbonated beverages, red wine, fish/seafood, legumes, nuts, commercial food and Mediterranean traditional dishes with tomato sauce) and the preferred cooking fat used and meat consumed. Each item was scored as zero (answer “no”) or one (answer “yes”) so that the sum resulted in a final score between 0–14. Additionally, the items for fruits, vegetables and “sweets, cookies and cake” were also evaluated individually. Fruit and vegetable intake below the recommendation of five portions per day was set as insufficient [77].

To assess drinking behaviour, self-developed questions were used. The participants were asked for their personal total daily liquid intake (Q: How much do you drink in total per day?) and how many glasses of caffeinated drinks they consume (Q: Do you drink beverages containing caffeine, such as coffee or coffee specialties, coke or energy drinks?; Q: How many cups of coffee or coffee specialties and glasses of coke or energy drinks do you drink a day?). Risky caffeine consumption was defined as 400 mg of caffeine per day [78], which equals five caffeinated drinks. Additionally, the degree of the work’s influence on the drinking behaviour was measured by a five-point Likert scale ranging from “no influence at all” to “very high influence” in response to the question *“Does your work influence a decreased drinking behaviour?”*. Furthermore, the participants were asked which underlying factors influence their drinking behaviour.

To measure physical activity, we used the validated German version of the 16-item Global Physical Activity Questionnaire (GPAQ) [79]. The GPAQ is a modified version of the International Physical Activity Questionnaire (IPAQ) and has also been validated against objective measures, such as accelerometry and pedometry [80,81,82,83]. The overall validity was fair to moderate and the highest correlations between GPAQ scores and accelerometer were found for sedentary time (*r* = 0.47) and vigorous physical activity (*r* = 0.46). Furthermore, a mostly negative difference between accelerometer and GPAQ data suggests that physical activity was overestimated in the GPAQ, with mean differences for total physical activity at M (SD) = −9.2 (16.0) hours/week [79]. In contrast to the IPAQ, the GPAQ differentiates between activities of moderate and intensive physical activity, and whether the activity was carried out during work, transport or leisure time. We used the official Analysis Guide of the World Health Organisation for evaluation [84]. The metabolic equivalents (MET) of physical activity were determined and based on this, the percentage distribution of physical activity between work, transport and leisure time was calculated. Moreover, the total physical activity and the physical activity in leisure time of outpatient caregivers was compared to the recommendation of the World Health Organisation [47] of ≥150 min physical activity per week. Ten participants were excluded from the evaluation of the GPAQ because they were either counted as outliers (*n* = 2) or their self-reported data lacked values for the time duration spent being physically active (*n* = 8).

To assess smoke exposure, we used the questionnaire by Latza et al. [85] and pack-years as well as smoker status were determined. In addition, we added questions on e-cigarette smoking status that matched the survey scheme of Latza, Hoffmann, Terschüren, Chang-Claude, Kreuzer, Schaffrath Rosario, Kropp, Stang, Ahrens and Lampert [85]. Nevertheless, no in-depth study of e-cigarette use was conducted, as the scales validated for e-cigarette consumption are currently only available in English [86].

Additionally, participants’ sleep quality was measured on a four-point scale ranging from “very poor” to “very good” in response to an item from the Pittsburgh Sleep Quality Index [87]: Q: “How would you rate the quality of your sleep over the past four weeks?”. A self-developed question asked whether it was easy for outpatient caregivers to keep their break times (Q: “Do you find it easy to keep your break times?”). Responses were given on a five-point ordinal scale ranging from “never” to “always”.

Lastly, the changes in eating behaviour (Q: “How would you rate your eating behaviour since the beginning of the COVID-19 pandemic?”), physical activity (Q: “How would you rate your physical activity since the beginning of the COVID-19 pandemic?”), smoking (Q: “How would you rate your smoking habits since the beginning of the COVID-19 pandemic?”), perceived stress (Q: “Since the beginning of the COVID-19 pandemic, my life has become more stressful than before!”) and sleep quality (Q: “How would you rate your sleep behaviour since the beginning of the COVID-19 pandemic?”) since the emergence of the COVID-19 pandemic were determined using self-developed questions on a five-point Likert scale. The response scales differ, but choosing the middle response in each scale means that there has been no change in the respective health behaviour since the occurrence of the COVID-19 pandemic. An overview of all variables used for the analysis as well as a specification of the self-developed items can be found in Appendix A.

#### 2.3.3. Statistical Analysis

Modes, percentages, means and standard deviations were calculated to report participants’ characteristics. Shapiro–Wilk tests showed that several variables were not normally distributed, so parametric test procedures were used for statistical analysis. Group comparisons were made using multiple Mann–Whitney U tests. The ordinal data regarding the difference in health behaviour during the COVID-19 pandemic were analysed by using multiple one-sample Wilcoxon Signed Rank tests. The test value 0 corresponds to the answer that the respective health behaviour did not differ before and during the pandemic. Significant results to a positive z-test statistic indicate that health behaviours (eating behaviour, physical activity, smoking behaviour and sleep quality) have improved, while negative z-test statistics indicate a deteriorated health behaviours since the beginning of the pandemic. For stress perception, the scale is inverted, so that higher values represent an increased perception of stress and lower values represent a lower perception of stress. Subsequently, the z-test statistics was used to compute Pearson’s standardised regression coefficient *r* [88]: r= z/√n. The effects were interpreted according to the very low (<0.2), low (0.2 < 0.4), moderate (0.4 < 0.6), strong (0.6 < 0.8) and very strong (≥0.8) as suggested by Bortz [89]. The significance level was set to α = 0.05. Statistical analysis was performed using IBM SPSS Statistics software for Windows (version 25.0, released 2017; IBM Corp., Armonk, NY, USA).

## 3. Results

In the following, the results of each approach, i.e., qualitative and quantitative, will be described in detail.

### 3.1. Qualitative Approach

#### 3.1.1. Sample Characteristics

As shown in Table 1, most of the interviewees were older than 30 years old. Of the 15 outpatient caregivers from Hamburg, Germany, three were male and 13 worked full-time with a work experience of over 5 years, so all of them have been working in the outpatient care for at least six months. Most of the 15 interviewees were qualified as geriatric nurses. Additionally, six of the interviewees had at least one child at home for whom they were responsible.

#### 3.1.2. Health Behaviour of Outpatient Caregivers

A general change in health behaviour among outpatient caregivers was shown in more caution in the daily life or intending to lose weight. Responses from outpatient caregivers regarding their health behaviour can be divided in eating behaviour, physical activity, drinking behaviour, smoking behaviour as well as behavioural patterns regarding breaks and regeneration and health-promoting behaviours.

Changes in eating behaviour varied between interviewed outpatient caregivers. The majority of the respondents reported no change in their eating patterns during the COVID-19 pandemic. There were some outpatient caregivers who reported a changed eating behaviour, i.e., better (in their own words). They talked about eating more fresh and home cooked meals since the beginning of the outbreak. Others described their eating behaviour as more negative due to the pandemic. Reasons were, for instance, the higher consumption of convenient food and ready meals. Some reported a generally reduced as well as increased food intake. However, this change happened regardless of the pandemic.


*“Usually I used fresh ingredients, now I have more frozen food because the joy of cooking has decreased due to this stress. I won’t stand in the kitchen for two hours, I’m glad if it’s only one.” (Outpatient Caregiver #3, 31–40 years, <1 year outpatient care experience)*


Looking at individual drinking behaviours, a major proportion of respondents reported no change at all due to the pandemic. Some pointed out an increase in drinking amount. However, most mentioned a decrease instead. A few highlighted a healthier selection of beverages in general. A popular choice among outpatient caregivers is coffee. Its consumption was unchanged; some emphasised that it was unwaveringly high. A few reported a higher coffee intake due to higher perception of stress.


*“No, it stayed the same, well I drink a lot of water and coffee and so on.” (Outpatient Caregiver #6, ≥41 years, 1–5 years outpatient care experience)*



*“Well at home I used to drink a lot more soft drinks such as Coke or Fanta but that’s reduced currently, significantly less” (Outpatient Caregiver #7, 20–30 years, 1–5 years outpatient care experience)*


A large proportion of surveyed outpatient caregivers pointed out that their physical activity in their leisure time has been diminished since the pandemic. According to the respondents, this was attributable to the restrictions of public life in general. Additionally, the first closure of gyms had a reducing effect on their physical activity in their free time because of new hindrances. Whereas a few reported the effect of closing gyms, others highlighted negative consequences, such as musculoskeletal complaints, due to the restricted possibility to work out. For some outpatient caregivers, a general reduction of physical exercise was due to their fear of contracting the coronavirus. Others only changed the venue and moved their exercises outdoors. However, a few respondents reported no change in their physical activity in their leisure time. In view of working time, the majority of participants reported no change in physical activity during their working activity. Some reported more movement, whereas others again even mentioned a decrease in movement during work.


*“Well, I used to go to the gym and that’s simply not possible anymore. You have to be creative at home but there are just not the same possibilities (…).” (Outpatient Caregiver #7, 20–30 years, 1–5 years outpatient care experience)*


Some of the interviewees were smokers. Most of the smoking outpatient caregivers reported no change in their amount of smoking cigarettes. Some of the respondents underlined their intention to quit smoking or to reduce it in quantity. Nevertheless, a couple of outpatient caregivers reported an increased smoking behaviour, while others complained about their mask smelling like cigarette smoke.


*“I am that person who likes to smoke while talking on the phone and that became more. Whereby actually, while I’m working I’m smoking less because I hate it when my mask is smelling like smoke.” (Outpatient Caregiver #11, ≥41 years, >5 years outpatient care experience)*


Most of the outpatient caregivers have experienced no change in their habits in terms of taking breaks. A couple more mentioned that taking breaks during work was different due to the obligation of keeping distance, contact minimisation as well as hygiene regulations. Some of the outpatient caregivers mentioned fewer breaks by reason of more work. Others mentioned difficulties because of the absent possibility of buying food outwardly.


*“Well, usually we liked to have breakfast together. But now this is over.” (Outpatient Caregiver #1, 31–40 years, <1 year outpatient care experience)*


When it comes to sleeping patterns, many outpatient caregivers reported no changes. However, a couple of outpatient caregivers complained about less sleep in general or developing sleep disorders, though a few even mentioned receiving more sleep since the beginning of the outbreak. In the view of regeneration, a lot of outpatient caregivers emphasised that opportunities of regeneration have decreased due to restrictions of public life or time pressure. Furthermore, most of the respondents highlighted current recreational opportunities available at home, such as using media or body care. Only a few mentioned no changes in their recreational opportunities, while others complained about less recovery because of childcare duties. Finally, some of the respondents mentioned that going for a walk or being in nature was experienced as regeneration.


*“Well, I can’t recover at all. Because of screaming children in the background when I want to relax.” (Outpatient Caregiver #8, 31–40 years, >5 years outpatient care experience)*


Health-promoting behaviour of outpatient caregivers at work was predominantly reflected by using personal protective equipment (PPE) at all times. In this context, the usage of mouth-nose protection, gowns and gloves was commonly mentioned.


*“Yes, we must protect ourselves with masks and gloves.” (Outpatient Caregiver #3, 31–40 years, <1 year outpatient care experience)*


Moreover, respondents often highlighted a higher hand hygiene in general. In this context, hand disinfectants to reduce germs and minimise the risk of infection were mentioned by interviewees.


*”Yes, one is washing hands more often suddenly. That’s truly the case. You wash your hands all the time (…)” (Outpatient Caregiver #2, 31–40 years, <1 year outpatient care experience)*


Increased hygiene measures also included the use of surface disinfectants, although participants mentioned they were not frequently used.


*“There is also an arrangement with the car that you clean the steering wheel, the gear stick, the door handle from the inside before you park the car again and the next person goes in there.” (Outpatient Caregiver #12, ≥41 years, >5 years outpatient care experience)*


The majority of outpatient caregivers also attached high importance to the generally recommended hygiene and distancing regulations.


*“I always try to be careful and do not become careless about respecting the distance rules (…).” (Outpatient Caregiver #14, ≥41 years, >5 years outpatient care experience)*


However, a few mentioned that surface disinfection took a lot of time, so due to this, time pressure was increased. Thus, the process was shortened by some outpatient caregivers.


*“I have to say that it was even more extreme in March, but now it’s not like you’re standing there all the time, then I’d be lying, because I don’t have the time to clean everything every two hours.” (Outpatient Caregiver #1, 31–40 years, <1 year outpatient care experience)*


Similar to their health-promoting behaviour in the workplace, the majority of the outpatient caregivers also used PPE in their private lives, i.e., for self-protection and the protection of fellow human beings. Masks were described as the main used source of equipment for the purpose of infection prevention.


*“Privately, I always wear mouth-nose protection.” (Outpatient Caregiver #5, 20–30 years, 1–5 years outpatient care experience)*


Increased hand hygiene was pursued in private as well. A high compliance with determined hygiene regulations were also observed individually.


*“Yes, we wash our hands much more often. Much more often and also [our children’s hands] much more often. It is very much in our consciousness.” (Outpatient Caregiver #8, 31–40 years, >5 years outpatient care experience)*


In addition, the outpatient caregivers also relied on the recommended social distancing rules in private in general. However, this was partly accompanied by psychological discomfort, e.g., due to a decrease in leisure activities or feelings of loneliness. Social distancing partly involved children too, as they could act as carriers of the virus to their own family and indirectly to patients as well.


*“In private? I have reduced my contacts, I do not go out privately.” (Outpatient Caregiver #10, 31–40 years, 1–5 years outpatient care experience)*



*“I try to take my child with me as little as possible.” (Outpatient Caregiver #3, 31–40 years, <1 year outpatient care experience)*


A few considered to use helping tools such as the Coronavirus Warning App (which had not been finalised yet at the time the interviews were conducted) in private to facilitate contact tracing in the future.


*“Yeah. That’s good. I think everything digital is great. (...). I just read a report that it works so well in China, so that would be something.” (Outpatient Caregiver #8, 31–40 years, >5 years outpatient care experience)*


Although the majority of those surveyed also relied on health-promoting measures to deal with COVID-19 in the private sphere, some stated that there had been no change in their health behaviour in general or mentioned health-promoting measures without reference to the pandemic only, e.g., the attendance of recommended preventive medical check-ups.


*“No, neither. I have already made sure to have plenty of fresh air beforehand, to go for walks and, yes, to do sports and all that. So I didn’t have the feeling that I had to do anything else during the pandemic (…) It didn’t affect my activity or anything else.” (Outpatient Caregiver #14, ≥41 years, >5 years outpatient care experience)*


### 3.2. Quantitative Approach

#### 3.2.1. Participants

The study sample consisted of 171 outpatient caregivers, of which 112 female (65.3%), 57 male (33.3%) and 2 diverse (1.2%). A total of 132 participants (77.2%) had German parents and 39 (22.8%) had a migration background. Additionally, 70 participants (40.9%) were overweight or obese. More detailed information on the demographic characteristics of the quantitative study sample is provided in Table 2.

#### 3.2.2. Health Behaviour of Participants

Eating Behaviour: The statistical analysis of the MEDAS resulted in a mean value of M (SD) = 6.27 (±2.2) out of a maximum of 14 points. In addition, only 30 participants (17.5%) stated to consume 5 servings of fruit and vegetables daily (3–4 servings: 54 participants (31.6%); 0–2 servings: 87 participants (50.9%)). For “sweets, cookies and cake”, 79 outpatient caregivers (46.2%) reported consuming 0 to 2 servings per week, 57 participants (33.4%) 3 to 4 servings and 35 participants (20.5%) 5 servings or more.

Drinking Behaviour: The outpatient caregivers reported an average daily fluid intake of M (SD) = 2.14 (±0.75) litres. In addition, 144 participants (84.2%) stated that they consumed caffeinated beverages and 80 participants (46.8%) drank 5 cups of caffeinated beverages or more a day. While 44 outpatient caregivers (25.7%) indicated that work had no negative influence on drinking behaviour at all, rarely any influence was reported by 28 participants (16.4%), some influence by 40 participants (23.4%), a high influence by 47 participants (27.5%) and a very high influence by 12 participants (7.0%). The most frequently named causes of influence by the outpatient caregivers were that they forgot to drink (35.7%), did not have enough time to drink (32.2%), did not want to use the toilet at the patient’s home (26.9%), no toilets were available (16.4%) and, lastly, that caregivers forgot to bring something to drink (10.5%). Three outpatient caregivers reported in the open comment box that at work, water was not available free of charge and water offered by patients should not be accepted according to company guidelines.

Physical Activity: Regarding the percentage distribution, the majority of the physical activity of outpatient caregivers (*n* = 161) took place at work (M (SD) = 50.85% (±33.59%)) compared to transport (M (SD) = 23.50% (±22.34%)) and leisure time (M (SD) = 25.66% (±27.35%)). From the self-reported data, 3 participants (1.9%) did not achieve 150 min of physical activity per week in total. Moreover, 60 participants (37.3%) reported less than 150 min of physical activity per week in leisure time, which was classified as inadequate physical activity based on recommendations of the WHO [47].

Smoking Behaviour: The sample included 80 non-smokers (46.8%), 36 former smokers (21.1%) and 55 current smokers (32.2%). Of the current smokers, 48 were tobacco smokers, 4 used e-cigarettes and 3 reported using both. The average tobacco consumption of ex-smokers and current tobacco smokers was M (SD) = 19.70 (±18.83) pack-years.

Regeneration Behaviour: The results of the query as to whether it is easy to take regular breaks varied widely. While 24 participants (14.0%) stated that it was never easy for them, this was rarely true for 35 participants (20.5%), sometimes for 44 participants (25.7%), often for 47 participants (27.5%) and always true for 21 participants (12.3%). In regard to the sleep quality, 105 outpatient caregivers (61.4%) reported their sleep quality to be good or very good, while 66 participants (38.6%) evaluated their sleep quality as inadequate (poor or very poor).

Figure 1 shows the proportion of outpatient caregivers from our sample who do not meet recommendations for the respective health behaviour and, thus, illustrates the salient reported results for: low fruit and vegetable intake, high caffeine consumption, overweight and obesity, inadequate sleep quality, inadequate physical activity and smoking status.

#### 3.2.3. Differences in Health Behaviour in Times of the COVID-19 Pandemic

Table 3 presents data comparing different health behaviours during the COVID-19 pandemic to before. The results show that the self-rated healthiness of eating behaviour (M (SD) = −0.19 (±0.85), *p* = 0.002, *r* = 0.22), quantity of physical activity (M (SD) = −0.37 (±0.87), *p* < 0.001, *r* = 0.42) and sleep quality (M (SD) = −0.65 (±0.80), *p* < 0.001, *r* = 0.81) was significantly decreased during the COVID-19 pandemic. Moreover, there was a significant increase in perceived stress (M (SD) = 0.41 (±1.22), *p* < 0.001, *r* = 0.34). The effect sizes were small for eating behaviour, physical activity and perceived stress and large for sleep quality.

#### 3.2.4. Subgroup Analyses

Analyses of subgroups revealed that male (m) outpatient caregivers showed a significantly lower adherence to the dietary quality score than their female (f) counterparts (M_m_ (SD) = 5.4 (±2.04) vs. M_f_ (SD) = 6.7 (±2.22), z = 4342, *p* < 0.001) and also reported an unhealthier eating behaviour since the onset of the COVID-19 pandemic (M_m_ (SD) = −0.39 (±0.92) vs. M_f_ (SD) = −0.07 (±0.78), z = 3852, *p* = 0.011). Female outpatient caregivers, however, experienced greater perceived stress than male outpatient caregivers since the onset of the COVID-10 pandemic (M_f_ (SD) = 58 (±1.17) vs. M_m_ (SD) = 0.07 (±1.27)), z = 3923, *p* = 0.012).

The comparison of age groups showed that more pack-years were reported among outpatient caregivers ≥40 years (≥40) of age (M_≥40_ (SD) = 22.9 (±19.87) vs. M_<40_ (SD) = 11.2 (±13.09), z = 1297.5, *p* = 0.002) and eating behaviours since the onset of the COVID-19 pandemic were rated as unhealthier among the age group of <40 years (<40) (M_≥40_ (SD) = 0.05 (±0.84) vs. M_<40_ (SD) = −0.51 (±0.75), z = 4693, *p* < 0.001).

Furthermore, overweight participants (O) showed significantly lower adherence to the dietary quality score than those of normal weight (N) (M_O_ (SD) = 5.9 (±2.43) vs. M_N_ (SD) = 6.6 (±2.03), z = 2795, *p* = 0.019). Even though significance was narrowly not reached for most values, overweight outpatient caregivers reported unhealthier eating behaviours (M_O_ (SD) = −0.30 (±1.03) vs. M_N_ (SD) = −0.11 (±0.69), z = 3058, *p* = 0.085), less physical activity (M_O_ (SD) = −0.51 (±0.91) vs. M_N_ (SD) = −0.27 (±0.84), z = 3022, *p* = 0.082), poorer smoking behaviour (M_O_ (SD) = 0.22 (±0.74) vs. M_N_ (SD) = −0.13 (±0.71), z = 465, *p* = 0.030) and higher perceived stress (M_O_ (SD) = 0.61 (±1.16) vs. M_N_ (SD) = 0.27 (±1.24), z = 4105, *p* = 0.066) than their normal-weight colleagues since the onset of the COVID-19 pandemic. More detailed information about the sub-group analyses can be found in Appendix A.

In the following, the hypotheses investigated are reviewed for support:

**Hypothesis** **1** **was** **accepted.***Outpatient caregivers rated the healthiness of their eating behaviour as significantly lower during the pandemic compared to before. However, sub-group analyses suggest that the low effect size found is mainly among male participants and those under 40 years of age*.

**Hypothesis** **2** **was** **accepted.***Outpatient caregivers rated their physical activity as significantly lower during the pandemic compared to before. The effect was moderate*.

**Hypothesis** **3** **was** **rejected.***Outpatient caregivers did not rate their tobacco use as significantly higher during the pandemic compared to before*.

**Hypothesis** **4** **was** **accepted.***Outpatient caregivers rated their perceived stress as significantly higher during the pandemic compared to before. However, sub-group analyses suggest that the low effect size found is stronger among female than male participants*.

**Hypothesis** **5** **was** **accepted.***Outpatient caregivers rated quality of their sleep as significantly lower during the pandemic compared to before. The effect size was very strong*.

## 4. Discussion

### 4.1. Discussion of the Interview and Survey Results

This is the first study applying qualitative and quantitative methods to examine outpatient caregivers’ health behaviour in relation to the COVID-19 pandemic. Important insights of their various health behavioural patterns related to the pandemic were gained. As outpatient caregivers have a special work setting, in which they are forced to work in the field and have direct contact with patients, the particular pandemic-related job demands could have a negative impact in terms of conducting health-promoting behaviours [68,90,91]. Surveyed outpatient caregivers—both in the qualitative and in the quantitative approach—reported different statements with regard to changes in their health behaviour patterns. Different tendencies of the results could be explained due to the comparatively smaller sample of the qualitative approach (cf. [92,93]). Nonetheless, quantitative survey results, which could be more difficult to understand, can be further explained by qualitative research results [94].

#### 4.1.1. Eating Behaviour

Overall, findings from the qualitative and quantitative approach are in close agreement. In the qualitative study, most respondents reported no change due to the COVID-19 pandemic, while only some interviewees mentioned a poorer eating behaviour due to higher ready meal intake, whereas the quantitative data showed small effects towards a deterioration of self-evaluated eating behaviour. A subgroup analysis showed that changes were present in male outpatient caregivers and those under 40 years old. In this context, survey results among nursing students show a weight gain due to the pandemic [52], which was shown among Brazilian urologists as well [95]. This could be explained by a higher stress perception among nurses in comparison to doctors (cf. [29]). The higher the stress perception, the higher the risk of conducting health-impairing behaviours [90]. In contrast, however, other results show that the pandemic combined with anxiety made it difficult for nursing students to eat regularly [96]. Nonetheless, Duong, Pham, Do, Kim, Dam, Le, Nguyen, Nguyen, Nguyen, Le, Do and Yang et al. [46] highlighted an improvement in the nutritional behaviour of nursing and medical students, which was achieved through the digital promotion of health literacy. Tran et al. also underlined the positive impact of higher health literacy on conducting more health-promoting behaviours, such as a more balanced diet among healthcare workers during the COVID-19 pandemic [54]. Frequent healthy eating was also recently associated with fewer depressive symptoms, lower anxiety and less perceived stress among Portuguese nurses during the COVID-19 pandemic [97]. Ultimately, the eating habits of outpatient caregivers seem to have been unfavourable even before the pandemic (irregular meal intake and high snacking) [36].

#### 4.1.2. Drinking Behaviour

In the interviews, most outpatient caregivers rarely reported changes regarding their drinking behaviour due to the pandemic, except for a partially higher coffee consumption due to higher perception of stress because of the COVID-19 pandemic. The results of the quantitative approach indicate that the total amount of drinking is sufficient, but the participants reported a negative impact on their drinking behaviour during work hours, resulting in forgetting to drink at all and a higher caffeine intake as well. Due to the timing of the survey, it is not possible to distinguish whether the changes are due to working conditions, the influence of the pandemic or both. Prior to the pandemic, however, it was already known that nurses and midwives tend to consume more coffee, especially if they work in shifts [98,99,100]. Since caffeine is known to reduce symptoms of tiredness [101] and job demands as well as stress perception are higher for outpatient caregivers during the COVID-19 pandemic [68], the desire for coffee consumption could be higher (cf. [102]). A few interviewees highlighted more positive beverage choices in the meantime. However, recently published study results show that female caregivers tend to choose rather unfavourable drinks (e.g., soda), especially in the case of increased job-related stress [103]. Moreover, some of the interviewees reported a decrease in general drinking amount, just like the quantitative survey participants. They either forgot to drink or they intentionally drank less because they did not want to use patients’ restrooms at home, which is often the only available opportunity while working. Challenges in terms of finding a restroom are already known among nurses and midwives [104]. Due to the lack of ventilation systems, airborne disease transmission through droplet infection is also increased in public toilets (cf. [105]). This could be even more limiting for outpatient caregivers during the pandemic. The problem of finding toilets could lead to outpatient caregivers neglecting adequate hydration, as their work environment is location-unspecific [106]. Further study results indicated a higher alcohol consumption during the COVID-19 pandemic among urologists [95] as well as Australian healthcare workers [107]; however, these results did not appear in the context of the present study. However, since health-damaging behaviour can be promoted by experienced work-related stress [90], this could also be a problem in outpatient care that is not openly stated by participants.

#### 4.1.3. Physical Activity

In both the qualitative and quantitative approach, outpatient caregivers indicated that they have been less physically active during the COVID-19 pandemic in their free time, although all of them reported sufficient physical activity at work. From the interviewees of this present study, the closure of the gyms and the everyday restrictions in public life turned out to be the main reasons. Recently published study results from health-related university students indicate the same tendency. The lockdown restrictions led to an overall decreased physical activity among participants [108]. Although Tran et al. [54] and Que et al. [109] highlighted a protective effect of sufficient physical activity against anxiety disorders and depressive symptoms through their study among healthcare workers, including inpatient nurses, the COVID-19 pandemic led to an observed decrease in physical activity in the inpatient health sector among healthcare workers, urologists and nursing students [52,95,108]. In contrast, other study results showed that in the inpatient setting, healthcare workers used physical activity as a coping behaviour in the course of the experienced job demands of the pandemic [110]. Portuguese nurses also showed increased physical activity during the COVID-19 pandemic [111].

#### 4.1.4. Smoking Behaviour

Smokers were among both the interviewees and the participants of the survey. At 32.2%, the smoking rate in our sample was above the average of the German population (28.1%) [44], and there was no change in smoking habits reported by outpatient caregivers due to the pandemic. A high smoking rate is also known from inpatient care, which is mostly associated with job-related stress perception, long working hours, shift work and duties towards family members [48,112,113]. However, smoking is highly prevalent as a health-impairing behaviour in the care sector compared to other healthcare workers [8,48,49,53,114]. Moreover, perceived stress levels can be increased in times of epidemics and, thus, during the current COVID-19 pandemic as well [115]. The first study results among outpatient caregivers regarding their job demands and stress perception, therefore, indicated that a decrease in tobacco amount would be difficult during the COVID-19 pandemic [68], especially because the initial findings on the health behaviour of outpatient caregivers from Germany before the outbreak of the coronavirus suggested that not only is the smoking rate high, but perceived work stress promotes smoking behaviour [36]. In the same course, study results on healthcare workers show that smoking may increase vulnerability to stress and anxiety disorders caused by the COVID-19 pandemic [54]. In another study among frontline nurses who smoked at the time of the survey, depressive symptoms were also more likely to be found [116].

#### 4.1.5. Regeneration at Work: Break Behaviour

Although a few interviewees mentioned no change in their break behaviour, they nonetheless had to design their rest breaks differently due to public restrictions (e.g., alone instead of with colleagues). Most of the outpatient caregivers who participated in the survey, however, stated that they would not be able to take their rest breaks at work during the COVID-19 pandemic, just like some of the interviewed outpatient caregivers. Reasons were, for instance, a higher workload due to the pandemic. The small chance of taking rest breaks is already known from the pre-COVID-19 context in the German care sector [117], even though rest breaks are legally defined in the German Working Hours Act (ArbZG) [118]. Known reasons for skipping breaks in care are particularly a high workload, time pressure, stress perception, shift work and understaffing (e.g., due to sick leave) [55,117,119,120]. Mo, Deng, Zhang, Lang, Liao, Wang, Qin and Huang [28] analysed stationary nursing staff from China. Fear, high workload and having kids caused stress perception, as reported by nurses. Work pressure during the COVID-19 pandemic is also shown by Zhang, Wei, Li, Pan, Wang, Li, Wu and Wei [34] examining frontline nurses. These factors also appeared among outpatient caregivers interviewed during the COVID-19 pandemic [68] or in the pre-corona context [36], which could also be a reason for participants of the present study not being able to take their rest break on a regular basis. Finally, it should be noted that smokers tend to take their breaks more often than non-smokers, which is known from other study results from both inpatient care [121,122] and outpatient care [36].

#### 4.1.6. Regeneration after Work: Sleep Behaviour

In both the qualitative and quantitative approach, participants reported lower sleep quality during the COVID-19 pandemic (large effect size of *r* = 0.81). Lockdown measures have been proven to have a negative impact on nursing students’ sleep quality [123]. Furthermore, sleep disturbances resulting from anxiety or depression due to the COVID-19 pandemic is already known among hospital nurses [58,61]. Moreover, perceived job stress among healthcare workers could be a predictor of poor sleep quality [124]. In addition, Zheng, Wang, Feng, Ye, Zhang and Fan [60] showed an association between shift work during the COVID-19 pandemic and poor sleeping patterns among medical care workers. The influence of shift work and shift type on sleep quality and quantity is also known from studies published before the COVID-19 pandemic on inpatient nurses and midwives (e.g., [57,100,125,126]). Sleep deprivation caused by shift work might also increase improvable food choices (e.g., snacking) [36,98,127]. Perceived work-related stress has also been shown to lead to poorer sleep quality among nurses working in shifts [125,126]. Since stress perception, anxiety disorders and depressive symptoms also occurred among outpatient caregivers during the COVID-19 pandemic [68], it can be assumed that there is also a possible association here between these negative strain reactions caused by the pandemic and impaired sleep quality of outpatient caregivers in our study. Interviewees of the present study, furthermore, mentioned fewer possibilities of regeneration in their leisure time due to public restrictions, which can be attributed to the imposed restriction of contact by the German government on 15 April 2020 to contain the COVID-19 pandemic initially [128]. Social isolation due to social distancing as well as impaired work-life balance is also mentioned in other studies [129,130].

#### 4.1.7. Stress Perception

Participants of the quantitative approach reported a higher stress perception during the COVID-19 pandemic. This is also in line with other study results examining outpatient caregivers [68]. The COVID-19 pandemic caused higher stress levels in the inpatient care setting [22,28,30,131,132,133,134,135], or burnout symptoms [136], which can be a result from a continuous stress perception [137]. Furthermore, in our quantitative study, perceived stress was higher among female than male participants. This can be explained by the multiple burdens that female caregivers face during the pandemic, e.g., additional care for their children at home due to day care and school closures [68]. Further study results also underlined that women are more likely to develop symptoms of mental disorders [20], such as depression, anxiety, insomnia and stress [22], and are consequently at higher risk of developing exhaustion and burnout syndromes [136]. Ultimately, a higher stress perception can make it difficult to carry out health-promoting behaviours and encourage health-impairing behaviours instead [90]. Recently published findings by Mojtahedzadeh, Rohwer, Neumann, Nienhaus, Augustin, Zyriax, Harth and Mache [36] further highlighted that outpatient caregivers perceive experienced work stress as a hindering factor for conducting health-promoting behaviours in general.

#### 4.1.8. Personal Health-Promoting Behaviour

Interviewees of the qualitative approach additionally reported on personal health-promoting behaviours during work, such as increased using PPE (wearing masks, gloves, and gowns), conducting more proper hand hygiene and keeping distance to their patients. Because of their special work setting in the field service and their obligation to visit patients in their homes, even though the German federal government recommended working from home whenever possible [91,128], outpatient caregivers may be at increased risks themselves; thus, precautions may be advised [68]. To control infection transmission, wearing PPE (masks, gloves, gowns and eye protection) is instructed [138]. As a matter of fact, it was strongly recommended by the Robert Koch-Institut [139] that employees in outpatient care wear masks permanently when working with their patients, who are usually risk groups themselves because of their age, comorbidities etc., to minimise the risk of infection at all times. An increased wearing of PPE, especially masks, has already been shown by other research findings when working in the care or midwifery setting and is classified under preventive behaviour [140,141,142]. This has also been displayed among medical and nursing students [143,144,145]. A general improved preventive behaviour was also shown among healthcare workers. The wearing of PPE intensified during work in the COVID-19 context [146]. Although the World Health Organisation [147] published an interim guidance on how to wear masks properly to prevent the transmission of the coronavirus, recent research findings have illustrated, however, that healthcare workers do not always wear masks in a correct manner [148]. Moreover, recent study results emphasise that wearing masks at work was perceived as stressful for outpatient caregivers as well as hospital healthcare workers, as they reported trouble breathing [68,130]. Furthermore, the correct usage of PPE in nursing professions requires sufficient knowledge of caregivers at all times (cf. [149]), as the awareness of it can improve the practice [150]. Proper hand hygiene, i.e., washing or disinfecting hands, is already widespread among healthcare workers (cf. [151]). Likewise, to ensure employees’ and patients’ safety, this is also internationally defined for healthcare professionals by the World Health Organisation and Patient Safety [151]. With regard to the COVID-19 pandemic, considering increased hand hygiene, sufficient performance was shown among healthcare workers, nurses and midwives [140,141,152,153].

In private, interviewed outpatient caregivers furthermore reported using PPE, particularly masks, in their private lives as well. In this context, the German federal government explicitly requested people to wear masks in daily life, especially while going grocery shopping or using public transportations [154]. Apart from this, interviewees also mentioned increased hand hygiene in private. This is also shown among Turkish adults, who showed improved personal preventive behaviours, such as washing hands more often and avoiding public transportation means, eating in restaurants and public gatherings [155]. Social distancing, which the interviewed outpatient caregivers named as another personal health-promoting behavioural pattern during the COVID-19 pandemic, ultimately led to feelings of loneliness for most of the interviewees. This challenge has been shown in former studies as well [129,130].

### 4.2. Strengths and Limitations

A particular strength of our study is its mixed-methods approach, providing both qualitative and quantitative data and expanding the current scientific evidence in relation to the COVID-19 pandemic. Moreover, we were able to incorporate the perceptions and views of outpatient caregivers of different ages, occupations and from different city districts of Northern Germany.

Looking at the qualitative approach, there are specific strengths, such as the precise orientation on recognised research methods, e.g., qualitative content analysis of collected data according to Mayring [70]. Moreover, a rigorous process of reflexivity and elaborate discussions on the findings and the use of direct quotes were made. Furthermore, participants of the qualitative study were recruited in a short time span from different outpatient care services of Northern Germany to get a picture of outpatient caregivers’ health behaviour during the COVID-19 pandemic, which has not been examined yet. However, potential limitations should be noted. Our findings are based on a random sample. Since participants were partly chosen via the snowballing technique, an increased risk of self-selection cannot be excluded. Furthermore, only three interviewees were male. The fact that female nurses seem to participate more often in studies should be mentioned [156,157,158], as well as the fact that more women work in care occupations in Germany [159]. Due to the COVID-19 pandemic, interviews had to be conducted as one-on-one telephone interviews instead of face-to-face, which resulted in a lack of eye contact and a distanced discussion atmosphere [160]. A resulting asynchronous communication in telephone conversations and reduced social cues are also a notable methodological limitation [161,162]. Another limitation can be the relatively small sample size. Ultimately, due to the qualitative research design, a generalisation of qualitative research results is not possible [92,163].

Specific strengths of the quantitative approach can be found in the use of validated instruments, except for drinking behaviour and diverse response formats, such as multiple choice as well as free-text fields in the survey. By using an online questionnaire tool, we were able to ensure that no data were missing. On the one hand, the quantification of results can represent the extent of the findings from the qualitative research and enables the comparability of health behaviour with the general population or other professional groups. On the other hand, our survey sample (*N* = 171) comprises roughly 0.04% of our target population (421, 550 outpatient caregivers working in German outpatient care services, cf. [164]), while the participation rate was low and drop-out rate was high. Hence, any judgement on the extent of representativeness is difficult. Nonetheless, as we have described above, there are no surveys investigating the health behaviour of German outpatient caregivers yet. Additionally, the results of our quantitative approach provide initial insights into the health behaviour of employees in the outpatient care. It should be noted that we further experienced great difficulties in recruiting eligible participants from the outpatient care in general. This could be due to the social distancing regulation due to the COVID-19 pandemic by the Federal German government [128]. We were told that outpatient care services were partially closed so that outpatient caregivers exclusively worked on the road. As described above, contact for recruitment had to be by telephone with the care managers, who wanted to get their staff to participate. Therefore, the success of the recruitment process was strongly dependent on both the care managers’ diligence and the caregivers’ self-motivation. In addition, new challenges as well as increased workloads due to the COVID-19 pandemic [117] led to exhaustion and higher time pressure among outpatient caregivers [68], which may also have reduced the willingness to participate in our survey. Moreover, the long survey period over several pandemic waves was necessary to generate more study participants, but may also have led to bias, as recent research suggests that prevention behaviour changes over time [165]. Different behavioural responses due to socio-demographic factors, increasing familiarity with prevention measures [165], as well as increased perceived psychological stress [166] due to the measures and the pandemic itself may bring about changes in health behaviour and may be confounding factors in relation to the survey timing. In addition, policy measures to contain the spread of the disease, such as mandatory wearing of face masks and closure of restaurants and shops, were implemented in Germany at different times, and thus, they were limiting participants’ health behaviour differently. Because there is no pre-post comparison, the data may be retrospectively biased and the validity of the results may be limited. Furthermore, the measurement instruments are subject to certain limitations. The use of the MEDAS in a non-Mediterranean context may lead to bias due to differences in other populations’ diets and lifestyles [167]. Nevertheless, the MEDAS was considered the most appropriate tool for our survey, as it has been validated for the German population and is the recommended survey tool for dietary quality in international guidelines [73,75]. In addition, the reliability of the GPAQ was assessed as good to very good, while the concurrent validity requires further investigation [168]. Nonetheless, the GPAQ has been validated for Germany and is widely used [79]. After all, self-assessments of health behaviour are not subject to objectivity and may additionally be distorted by social desirability [169].

### 4.3. Practical Implications for Further Research and Practice

Further research studies with larger sample sizes are needed in the future, e.g., in the subgroup analyses of normal-weight and overweight participants on the change in health behaviour during the pandemic, tendencies are apparent that did not reach statistical significance as the sample was too small. Moreover, outpatient caregivers display a special group of employees, high in diversity, who have a specific mobile work environment and their job demand during the COVID-19 pandemic could be higher [68]. In such studies, it would be interesting to conduct interviews with outpatient caregivers who have experienced the COVID-19 pandemic for over a year now and how their health behaviour might have changed over time. In the meantime, various vaccines are available to protect against infection with the coronavirus, and outpatient caregivers were part of the prioritised employee group in Germany in the first quarter of 2021 due to the Corona Vaccination Ordinance [170,171]. It would, therefore, be of research interest to investigate the status quo of vaccinated outpatient caregivers and their health behaviour (vaccinated vs. non-vaccinated) during the COVID-19 pandemic. Finally, it could be of future research interest to not only expand the sample size but also to achieve a more representative study sample, e.g., characteristics which should be considered could be different ages and an even gender distribution, using a quantitative questionnaire study or in relation with qualitative interviews to conduct another mixed-methods study. After more research has been carried out, specific interventions within the framework of work-side health promotion and occupational health and safety could be developed and implemented as they could have a positive effect on outpatient caregivers’ health behaviour (cf. [106,172]).

The implications for practice could be divided in behavioural (improving coping competences) and structural prevention (changes in the work organisation and environment) [173,174,175]. Outpatient caregivers should be educated about a healthy diet and nutrition (cf. [175]). In this context, a higher health literacy could imply a better health behaviour by carrying out knowledge in practice [176]. Therefore, behavioural interventions to increase outpatient caregivers’ health literacy should be improved (cf. [54]). Because of the fact that overweight and obese people are at more risk for more severe COVID-19 [177,178,179,180], educational interventions regarding better nutrition might be even of more relevance. For those outpatient caregivers consuming caffeine and tobacco on a daily basis, it might be useful to educate those regarding negative side effects and teach them about alternative coping strategies instead. For instance, coffee can lead to increased anxiety and impaired sleep [101], whereas smoking is followed, e.g., by an increased risk of cancer [181]. In this way, consumption amount could be better controlled even in times of an ongoing pandemic (cf. [182]). Educating outpatient caregivers regarding physical activity is also advised (cf. [183]). Furthermore, outpatient caregivers should be enlightened about how important it is to take breaks as it can enhance health and productivity [184,185,186]. In addition, strengthening personal resources could decrease stress perception and resulted in impaired sleep quality, favoured by shift work [187,188]. Improving outpatient caregivers’ resilience in general could lead to a decreased perception of stress. Therefore, trainings to strengthen personal resources are overall recommended [175]. However, behavioural interventions seem to be more difficult to implement in practice (cf. [189]). Hence, structural prevention measures become more relevant [190].

On a structural level, several things could be done to ensure healthy working conditions despite the COVID-19 pandemic. Regarding outpatient caregivers’ eating behaviour, it could be mentioned that their mobile work setting and constant time pressure, which is even higher during the pandemic, could make healthy eating more difficult [68,106,191]. The lack of toilet facilities, which is aggravated in pandemic times with infectious diseases [105], could lead to less liquid supply (cf. [104]). Therefore, employers in the outpatient care setting should ensure that their employees have sufficient access to disinfected toilet facilities during their work shifts, which could only be used by them and would be properly cleaned after each use (cf. [192]). A high workload and shift work should be avoided in the future to prevent outpatient caregivers from feelings of exhaustion, so that they have enough energy left to still be physically active after work (cf. [113,193,194,195,196]). Overall, workplace health promotion offers, e.g., in the form of programmes to promote resilience, and information sessions, e.g., on the topic of smoking, should be offered by the employer [197] or, for instance, offers on yoga to decrease stress perception [198]. Smoking cessation programmes also seem highly relevant during the pandemic. Recently published study results have revealed that smokers generally may have a more severe COVID-19 course [199,200,201]. Moreover, employers do not only have to understand the positive impact of regular breaks during work [186], breaks need to be scheduled enough for them on an organisational level. High workload, time pressure, stress perception and understaffing should be prevented in the future, as these are factors for not taking breaks regularly (cf. [55,117,119,120]) and are already occurring in the care context during the COVID-19 pandemic [28,34]. Furthermore, given the positive effects of social support [202], team spirit, communication and social exchange between colleagues and superiors should be ensured despite the ongoing COVID-19 pandemic to decrease feelings of loneliness [203,204]. With regard to preventive behaviours, potential negative physical strain reactions, such as trouble breathing [68], could be ensured by employers in outpatient care by, for instance, planning each tour of outpatient caregiver with enough possible rest breaks until they have to get to the next patient (cf. [205]). Nonetheless, a target group-specific development remains indispensable [172,206], especially considering the fact of the constantly ageing employees in outpatient care [164]. All in all, healthy working conditions could not only encourage health-promoting behaviour of outpatient caregivers, but also improve their general health, motivation and productivity [106], which seems to be even more important during the COVID-19 pandemic [67,68].

## 5. Conclusions

The present mixed-methods study focused on the health behaviour of outpatient caregivers during the COVID-19 pandemic, a yet unexplored field. Our findings demonstrate an initial insight into the individual health behaviour of outpatient caregivers during the COVID-19 pandemic. Thus, our study contributes to gaining a deeper understanding about caregivers’ self-understanding of their health behaviour. Subjective perceived stress or higher workload due to the pandemic could inhibit conducting health-promoting behaviours. However, our findings need further support by longitudinal studies which examine actual changes in health behaviour over time. Nevertheless, creating health-promoting working conditions and strengthening personal resources could help implement more health-promoting behaviours (cf. [90,97,106,111]). Given our finding that some behavioural patterns have become more health-impairing since the beginning of the pandemic, these aspects should be taken into further consideration in the planning of a better work environment in outpatient care services during the COVID-19 pandemic.

## Figures and Tables

**Figure 1 ijerph-18-08213-f001:**
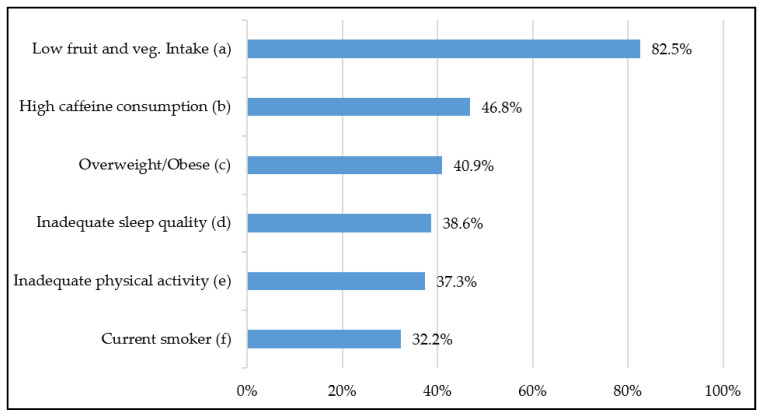
Proportion of the study sample (*N* = 171; for inadequate physical activity: *n* = 161) that did not meet the recommendations for aspects of the respective health behaviours. (a) Low fruit and vegetable intake was defined as <5 portions a day [77]. (b) High caffeine consumption was defined as ≥5 cups (200 mL) a day [78]. (c) Overweight and obesity were defined as ≥25.0 kg/m^2^ and ≥30.0 kg/m^2^, respectively [71]. (d) Inadequate sleep quality was determined based on the responses “very poor” and “poor” to the question “How would you rate the quality of your sleep over the past four weeks?”. (e) Inadequate physical activity was defined as <150 min of physical activity per week in leisure time [47]. (f) Current smoker was defined through the questionnaire used as smoking at least 5 cigarettes/cigarillos or 2 cigars/pipes [85].

**Table 1 ijerph-18-08213-t001:** Sample characteristics of interviewees (*N* = 15).

Sample Characteristics	*n*
Gender	
Female	12
Male	3
Age (years)	
20–30	2
31–40	7
≥41	6
Children in household	
0	9
≥1	6
Date of Interview	
May 2020	9
June 2020	6
Qualification	
Caregiver	1
Geriatric nurse	8
Home and family care	1
Healthcare and nursing staff	1
Social manager	1
Geriatric nurse, additional qualification intensive and palliative care	1
Wound expert	1
Geriatric nurse and wound expert	1
Occupation	
Outpatient geriatric nurse	8
Outpatient home and family caregiver	1
Outpatient caregiver	2
Outpatient geriatric nurse and office manager in health sector	1
Care specialist and nutrition manager in the outpatient care	1
Care specialist and deputy care management in the outpatient care	1
Outpatient geriatric nurse and wound expert	1
Work experience (years)	
<1	2
1–5	5
>5	8
Work Schedule	
Full-time	13
Part-time	2

**Table 2 ijerph-18-08213-t002:** Demographic characteristics of the study sample (*N* = 171).

Sample Characteristics	*n* (%)
Gender	
Male	57 (33.3%)
Female	112 (65.5%)
Diverse	2 (1.2%)
Age (years)	
18–29	18 (10.5%)
30–39	55 (32.2%)
40–49	40 (23.4%)
50–59	44 (25.7%)
≥60	14 (8.2%)
Origin	
German Parents	132 (77.2%)
Migration Background	39 (22.8%)
Highest Education Level	
General Secondary School	18 (10.5%)
Intermediate Secondary School	87 (50.9%)
Specialised Grammer School	21 (12.3%)
Grammar School	45 (26.3%)
BMI	
Normal Weight	101 (59.1 %)
Overweight	40 (23.4 %)
Obese	30 (17.5 %)

**Table 3 ijerph-18-08213-t003:** Outpatient caregivers’ self-evaluated change in health behaviour during the COVID-19 pandemic.

Health Behaviour	M	SD	Z ^a^	*p*	r ^b^	95% CI
Eating Behaviour	−0.19	0.85	−2.89	**0.004**	0.22	[−0.32, −0.06]
Physical Activity	−0.37	0.87	−5.51	**<0.001**	0.42	[−0.50, −0.24]
Smoking	0.02	0.73	0.18	0.855	-	[−0.09, 0.13]
Perceived Stress	0.41	1.22	4.40	**<0.001**	0.34	[0.23, 0.59]
Sleep Quality	−0.65	0.80	−10.60	**<0.001**	0.81	[−0.77, −0.53]

Note. *N* = 171. ^a^ One-sample Wilcoxon Signed Rank tests (test value = 0). Eating behaviour: I currently eat much unhealthier (−2)/unhealthier (−1)/just as healthy (0)/healthier (+1)/much healthier (+2) than before the COVID-19 pandemic. Physical activity: I am currently physically active much less often (−2)/less often (−1)/the same (0)/more often (+1)/much more often (+2) than before the COVID-19 pandemic. Smoking behaviour: I currently smoke much less often (−2)/less often (–1)/neither less often nor more often (0)/more often (+1)/much more often (+2) than before the COVID-19 pandemic. Perceived stress: My life has become more stressful since the start of the COVID-19 pandemic than before!—I do not agree at all (−2)/disagree (−1)/neither agree nor disagree (0)/agree (+1)/agree completely (+2). Sleep quality: I currently sleep much worse (−2)/worse (−1)/neither worse nor better (0)/better (+1)/much better (+2) than before the COVID-19 pandemic. ^b^ *p* < 0.05 in bold. Regression coefficients r were used as effect size. Effect sizes were categorised according to Bortz [89]: very low (<0.2), low (0.2 < 0.4), moderate (0.4 < 0.6), strong (0.6 < 0.8) and very strong (≥0.8).

## Data Availability

The data analysed during the current study are not publicly available due to German national data protection regulation. They are available on individual request from the corresponding author.

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
