# Peer review of "The Health Behaviour of German Outpatient Caregivers in Relation to the COVID-19 Pandemic: A Mixed-Methods Study"

_ijerph, 2021, doi:10.3390/ijerph18158213_

Round 1

Reviewer 1 Report

I have very grateful to revise this manuscript.

This is an interesting study that used a mixed-methods to study the health behaviour of outpatient caregivers during the COVID-19 pandemic.

The authors note that there are no studies that evaluate caregivers' health behaviors in the pandemic context. However I recommend that you read these articles: https://doi.org/10.1016/j.envres.2021.110828 and https://doi.org/10.3390/ijerph18073490 These manuscripts address health behaviors (healthy eating, physical exercise, etc) in Portuguese nurses. It might be interesting to support your conclusions.

Results

Table 2 - Authors should confirm whether they really mean “Sex” or whether they mean “Gender”.

Discussion

I don’t understand this phrese: “They either forgot to drink or they chose not to use patients’ restrooms at home.” What does the use of restrooms have to do with healthy drinking?

Conclusion

The authors should not claim either “Our findings demonstrate a high differentiation of insight into the individual experienced and implemented health behaviour of outpatient caregivers during the COVID-19 pandemic”. The fact that participants feel that they have improved or worsened with respect to certain health behaviors does not mean that this is real. To state this would require a longitudinal study. The authors should rephrase this.

Author Response

This is an interesting study that used a mixed-methods to study the health behaviour of outpatient caregivers during the COVID-19 pandemic.

The authors note that there are no studies that evaluate caregivers' health behaviors in the pandemic context. However I recommend that you read these articles: https://doi.org/10.1016/j.envres.2021.110828 and https://doi.org/10.3390/ijerph18073490 These manuscripts address health behaviors (healthy eating, physical exercise, etc) in Portuguese nurses. It might be interesting to support your conclusions.

Thank you very much for your appreciation! We looked into your recommended studies and included them in the discussion section.

Results

Table 2 - Authors should confirm whether they really mean “Sex” or whether they mean “Gender”.

Thank you for this attentive remark. As in Table 1, we actually meant and thus changed it to gender in Table 2 as well.

Discussion

I don’t understand this phrese: “They either forgot to drink or they chose not to use patients’ restrooms at home.” What does the use of restrooms have to do with healthy drinking?

Thank you for your comment. We meant that outpatient caregivers implied they either forgot to drink at all or they did not drink (enough or when they were thirsty) because if they drank more and needed to go to a restroom, the only possibility would be to use the patients’ private restroom which they wanted to avoid or were not allowed to do. We added some more explanation to the sentence to enhance its comprehensibility.

Conclusion

The authors should not claim either “Our findings demonstrate a high differentiation of insight into the individual experienced and implemented health behaviour of outpatient caregivers during the COVID-19 pandemic”. The fact that participants feel that they have improved or worsened with respect to certain health behaviors does not mean that this is real. To state this would require a longitudinal study. The authors should rephrase this.

Thank you very much for your comment. We considered it and rephrased our sentence.

Reviewer 2 Report

The subject of study of this paper is very timely given the current pandemic situation and the increase in Outpatient Caregivers in Germany. The introduction manages to bring us closer to the subject of study adequately.
The use of a mixed methodology is very appropriate.
The qualitative results are very superficial, depth is not reflected in the content analysis or in the motivations for the behaviors of the interviewees. You need to dig deeper into the analysis.
As the authors themselves point out, the quantitative sample is scarce to be able to generalize the results. In my opinion, the results obtained are poor and insignificant and do not justify such broad implications for future practice. The implications for employers presented are applicable regardless of the existence of a pandemic situation such as the current one or not. I suggest that the analysis of the qualitative results be improved by focusing on the motivations for the changes in behavior to achieve more specific conclusions on the subject of study raised and that they are novel.  

Author Response

Answer: Thank you very much for your feedback. We appreciate your remarks on the qualitative design. We would however like to kindly point out that our aim was to explore and gain first insights in a mostly unexplored research area. The subject area “motivations for the changes in behavior” was not in the foreground of the study objective. In order to keep the manuscript’s length comprehensible and readable and due to the combination of qualitative and quantitative methods and results, we presented the key findings of our qualitative study. Our intention was to support them with our quantitative findings to increase their significance and relevance. Although we appreciate your comment, unfortunately we cannot change our data, but will keep this in mind for further research.

Reviewer 3 Report

This is a study focused on health behaviour and the changes perceived among outpatient caregivers due to the COVID-19 pandemic. It is an interesting study that has clear limitations stated by the authors. However, it does fulfil its objective. The authors should respond to some minor questions outlined below. In general terms, the study is excessively long, and requires summarisation of certain sections (for example, the limitations or implications for practice). Title: appropriate. Introduction: The concepts of health behaviour and outpatient caregivers need to be explained more clearly at the beginning of the study. Who are the employees of outpatient service providers? Description of the healthcare system in Germany could be of interest. Background: The authors offer a profuse description of the problem. Health behaviour is divided into eating, drinking, physical activity, tobacco consumption behaviour and regeneration behaviour. This section is too extensive. A further description of outpatient care in Germany, its functions, competencies and the sector itself must be made in order to improve international readers’ understanding. Objective: well-stated. The research question is correct as well as the hypothesis outlined. Methods: Why was the qualitative study carried out first? The qualitative study design is not clear. Do the authors mean a Qualitative descriptive Study? Telephone interviews cause valuable interaction data to be lost by the interviewer, which makes data saturation difficult. The authors sent emails to a significant sample taken from outpatient care, please explain this concept. How many interviews were performed on each participant? Were any repeated? Why was participants’ pre-understanding excluded? Regarding the quantitative study, clarify the design. Why were only approximately 1/3 of the surveys completed? Although validated instruments are used, the evaluation of drinking behaviour is measured using your own non-validated questionnaire, therefore, what consequences might this have on the study? The representativeness of the sample, as acknowledged by the authors in the limitations, is questionable. Results:  interesting, pleasant to read, with a markedly descriptive nature.  Good discussion of results and extensive explanation of the study’s limitations. Conclusion: should respond clearly to the objectives. Ample implications for practice that need to be summarised. Bibliography is relevant, but overly extensive.

Author Response

Answer: 

Thank you very much for your thoughtful comments! We added some short explanation to the outpatient care sector in Germany in order not to make it too extensive. We also shortened the Background section. We hope it improves the international readers’ understanding. Both studies started in May 2020 in an explorative approach to collect data on the health behaviour of outpatient caregivers in Germany. Especially the interviews which were conducted during the first COVID-19-related wave and lockdown followed this exploratory approach. We intended to use the quantitative data of a larger sample to support our exploratory findings. Our aim was not to be representative, but to illustrate the in-depth views of selected outpatient caregivers to provide a starting point for further, quantitative research (cf. Kruse, 2015). Moreover, instead of face-to-face interviews, telephone interviews had to be carried out due to the pandemic and the lockdown. We are aware that we therefore lost valuable interaction cues, however it was not possible to conduct the interviews face-to-face. Therefore, we discussed this limitation in section 4.2. Given the exploratory approach we did not perceive this would make data saturation difficult, but will keep this remark in mind for future studies. The interviewees were interviewed only once and none of the interviews were repeated, we added this in section 2.2.1. We sent emails to eligible outpatient care services to recruit participants based on the gatekeeper principle. Thus, the outpatient care service managers could forward the information on our study to their employees (cf. Helfferich 2011).

In addition, there must be a misunderstanding here, no participants’ pre-understanding was excluded.

Regarding the quantitative study, we used a cross-sectional design and deployed an online questionnaire to gather data. We added this to section 2.3.1. in the first sentence. In quantitative surveys, the response rate often lies between 5-40% (Döring & Bortz 2016). Especially with online questionnaires, many interested persons might access the survey homepage without intending to complete it. Although we used self-developed questions to measure drinking behaviour, we are confident that they do not entail specific limitations since we directly asked for the amount of daily liquid intake. Also, we modified our conclusion with regard to our study aims.

References

Döring, N. & Bortz, J. (2016). Forschungsmethoden und Evaluation in den Sozial- und Humanwissenschaften [Research methods and evaluation in the social and human sciences] (5th ed.). Berlin: Springer.

Helfferich, C. (2011). Die Qualität qualitativer Daten. Manual für die Durchführung qualitativer Interviews [The quality of qualitative data. Manual for conducting qualitative interviews] (4 ed.). Wiesbaden: VS Verlag für Sozialwissenschaften.

Kruse, J. (2015). Qualitative Interviewforschung. Ein integrativer Ansatz. Mit Gastkapiteln von Christian Schmieder, Kristina Maria Weber sowie Thorsten Dresing und Thorsten Pehl [Qualitative Interview Research. An integrative approach. With guest chapters by Christian Schmieder, Kristina Maria Weber as well as Thorsten Dresing and Thorsten Pehl] (2nd ed.). Weinheim: Beltz Juventa.

Round 2

Reviewer 2 Report

I understand your arguments although I still think that your paper would improve with my suggestions.

Author Response

Dear reviewer,

the constructive advice has substantially improved our paper. We have modified the manuscript thoroughly with regard to the reviewer's suggestions.